# Repurposing caspofungin as a small-molecule inhibitor of *Clostridium perfringens* α-toxin for treatment of gas gangrene

Masaya Takehara [1,2] ✉, Yuta Homma[1], Tomoaki Ishihara [3], Yoshihiko Sakaguchi[2], Yusuke Kasai[4], Kanako Matsumoto[2], Katsuyuki Nakashima[5], Toshiyuki Yamaji[1], Yoshiyuki Tanaka [5], Hiroshi Imagawa[4] & Masahiro Nagahama[2]

## Abstract

**Background** Among pharmaceuticals currently in clinical use, few drugs directly target bacterial toxins. *Clostridium perfringens* α-toxin, a phospholipase C (PLC), is a major virulence factor responsible for gas gangrene caused by *C. perfringens* type A. There is a clinical need for small-molecule compounds that inhibit such bacterial toxins.

**Methods** A library of 764 FDA-approved drugs was screened to identify compounds that inhibit the PLC activity of *C. perfringens* α-toxin. Identified hits were further evaluated for their ability to inhibit α-toxin-induced cytotoxicity in human umbilical vein endothelial cells (HUVECs). Additional in vitro assays were conducted to assess changes in neutrophil activation and cytokine production. In vivo efficacy was evaluated in female C57BL/6J mice (n = 21 or 18 per group) challenged with purified α-toxin or infected with *C. perfringens* type A.

**Results** The initial screen identifies 21 compounds that inhibit the PLC activity. Among them, micafungin, an antifungal agent, is the only compound that suppresses α-toxin-induced cell death in HUVECs. Micafungin also reduces α-toxin-induced CD11b expression in neutrophils and cytokine release in HUVECs. Caspofungin, another antifungal with similar properties, also inhibits α-toxin-induced cell death and cytokine production. In mouse models, caspofungin, but not micafungin, significantly reduces lethality caused by α-toxin. Caspofungin also improves survival and mitigates muscle damage in mice infected with *C. perfringens* type A.

**Conclusions** Caspofungin demonstrates promising therapeutic potential as a life-saving treatment for gas gangrene caused by *C. perfringens* type A, likely through its inhibitory action on α-toxin activity. These findings support the development of new classes of small-molecule therapeutics that directly target bacterial toxins.

## Plain Language Summary

Gas gangrene is a life-threatening infection caused by a type of bacteria called *Clostridium perfringens*. This bacterium releases a harmful toxin called α-toxin, which destroys tissues and can lead to rapid disease progression and death. In this study, we tested 764 drugs that are already approved for use in humans to see if any could block the damaging effects of α-toxin, and found that two antifungal drugs, micafungin and caspofungin, were able to protect human cells from the toxin. Among them, caspofungin improved survival and reduced muscle damage in mice infected with *C. perfringens*. These findings suggest that caspofungin, a drug already used to treat fungal infections, could potentially be repurposed as a new treatment for gas gangrene. This offers hope for faster and more effective ways to treat this dangerous infection using existing medications.

---

*Clostridium perfringens* type A is a spore-forming, anaerobic Gram-positive bacterium that causes gas gangrene after traumatic injury[1,2]. This disease is a highly lethal necrotizing soft tissue infection characterized by the rapid development of myonecrosis, shock, and multiple organ failure[3]. Spontaneous non-traumatic *C. perfringens* sepsis occasionally occurs, develops rapidly, is accompanied by intravascular hemolysis and metabolic acidosis,

and has a high mortality rate[4]. The principal therapeutic measures are hyperbaric oxygen therapy, the administration of antibiotics, and surgical drainage; however, these treatments do not always prevent disease progression[3].

*C. perfringens* type A produces a number of virulence factors containing exotoxins, such as α-toxin (or phospholipase C) and θ-toxin (or

---

perfringolysin O[2]. Of these virulence factors, α-toxin has been identified as the main causative agent of gas gangrene. Using a genetically manipulated strain of *C. perfringens*, signs of fulminant muscle necrosis were reduced in mice infected with an α-toxin-deficient strain[5]. The relationship between the biological activity of α-toxin, which exhibits both phospholipase C (PLC) and sphingomyelinase (SMase) activities, and the progression of gas gangrene has been attracting increasing attention[2,6]. α-Toxin promotes the formation of platelet-leukocyte aggregates, which impair the host immune response by preventing the migration of leukocytes to the infection site[7–9]. These aggregates are formed via the activation of gpIIb/IIIa in platelets and the up-regulated expression of CD11b/CD18 in the cell membranes of leukocytes[10,11]. We previously reported that α-toxin inhibited neutrophil differentiation, leading to the impaired production of mature neutrophils and dysfunctions in innate host immunity[12,13]. These findings demonstrated the involvement of α-toxin in immune evasion by bacteria. The α-toxin-mediated formation of platelet-leukocyte aggregates has also been shown to induce intravascular occlusion[14,15]. Additionally, α-toxin caused endothelial cell death, which accelerated disturbances in the peripheral circulation[16]. Moreover, α-toxin inhibited erythroid differentiation, which may lead to erythrocytopenia and promote ischemia[17]. Collectively, these findings suggest that α-toxin causes ischemia in muscle tissue by impairing the peripheral circulation. The regeneration of muscle tissue from necrosis caused by *C. perfringens* was previously shown to be very slow[18]. We recently reported that α-toxin impaired skeletal muscle differentiation, which may have a negative impact on the repair of muscle tissue[19]. Therefore, α-toxin plays a role in disease development through various mechanisms, and the infection process has become increasingly clear.

In the present study, we screen a library of 764 Food and Drug Administration (FDA)-approved drugs and identify 21 compounds that inhibit the PLC activity of α-toxin. Subsequent studies show that micafungin and caspofungin, antifungal agents, reduce α-toxin-induced activities. We herein demonstrate the potential of caspofungin as a treatment for *C. perfringens*-induced gas gangrene due to its suppression of α-toxin, which provides insights for the development of therapeutic agents for infectious diseases caused by bacterial toxins.

## Methods

### Mice

Specific pathogen-free female C57BL/6J mice were purchased from Japan SLC, Inc. (Shizuoka, Japan) and housed under clean conditions with free access to food and water. Female mice were used to avoid unintended infections caused by male aggression and to prevent potential effects on survival outcomes, thereby ensuring reliable and consistent experimental results. Animals purchased for the study were randomly assigned to cages, with 3–5 mice per cage. No specific measures were taken to control potential confounders such as treatment order, measurement order, or cage location. Mice older than 8 weeks were used in all experiments. Animal experiments were approved by the Animal Care and Use Committee of Tokushima Bunri University (Approval No. 19-5), and procedures were performed in accordance with institutional guidelines. Institutional guidelines conformed to the Fundamental Guidelines for Proper Conduct of Animal Experiment and Related Activities in Academic Research Institutions under the jurisdiction of the Ministry of Education, Culture, Sports, Science and Technology, 2006.

This study investigates the severity of *C. perfringens* type A infection and the effects of its α-toxin in the host. Mice were used as a model because they are a well-established system for studying bacterial infections, and findings from this model are relevant for understanding aspects of human disease. No study protocol was prepared prior to the experiments.

### Reagents and strain

The Amplex red phosphatidylcholine-specific phospholipase C assay kit and Amplex red sphingomyelinase assay kit were purchased from Thermo Fisher Scientific Inc. (MA, USA). Clindamycin was obtained from Tokyo Chemical Industry Co., Ltd. (Tokyo, Japan). Micafungin, caspofungin, and peptidoglycan from *Bacillus subtilis* were purchased from Sigma-Aldrich

(MO, USA). Cell Counting Kit-8 (CCK-8) and Cytotoxicity LDH Assay Kit-WST were from Dojindo Molecular Technologies, Inc. (Kumamoto, Japan). Fluorescein isothiocyanate- or phycoerythrin-conjugated specific antibodies against mouse CD11b (clone M1/70, catalog no. 557396, lot no. 4310683) or Ly-6G/6C (Gr-1; clone RB6-8C5, catalog no. 561084, lot no. 3312623) and purified rat anti-mouse CD16/CD32 (Fc Block; clone 2.4G2, catalog no. 553141, lot no. 6092754) were purchased from BD Biosciences (CA, USA). The SCREEN-WELL FDA-Approved Drug Library V2, Japan version was obtained from Enzo Life Sciences, Inc. (NY, USA). All other chemicals were of the highest grade available from commercial sources. Strain 13 was used as wild-type *C. perfringens* type A strain.

### Drug screening methodology

The effects of primarily 50 μM of the test compounds (only one at 10 μM) on the PLC activity of 50 ng/ml α-toxin were measured using the Amplex red phosphatidylcholine-specific phospholipase C assay kit according to the manufacturer's instructions. After one hour of incubation, the measurement was performed by detecting fluorescence intensity with excitation at 550 nm and emission at 580 nm.

The effects of micafungin or caspofungin on SMase activity were measured using Amplex red sphingomyelinase assay kit according to the manufacturer's instructions. The measurement was performed by detecting fluorescence intensity with excitation at 550 nm and emission at 580 nm.

### Purification of α-toxin

α-Toxin was purified as previously described[12]. Briefly, *B. subtilis* ISW1214 was transformed with recombinant forms of pHY300PLK, which harbors the structural α-toxin gene, and was then cultured in Luria-Bertani broth supplemented with 50 μg/ml ampicillin at 37 °C. The culture medium containing secreted α-toxin was collected, and α-toxin was chromatographically purified.

The lethal activity of α-toxin in mice was measured using the following method. C57BL/6J mice were injected intraperitoneally with 600 ng of α-toxin and 300 μg of micafungin or caspofungin, which had been pre-mixed and diluted in phosphate-buffered saline (PBS), and the survival of mice was monitored. The experimental unit was a cage of animals. The exact number of mice per group and the total number of mice per experiment are provided in the figure and figure legend. No a priori sample size calculation was performed. Sample sizes were chosen based on previous experience with similar experiments and expected variability. No specific exclusion criteria were established, and no data points or animals were excluded from the analysis. The experiment was conducted by investigators who were aware of the group allocations, while outcome assessment was performed by a separate evaluator who was blinded to the group assignments. Humane endpoints were established such that animals exhibiting more than 20% body weight loss compared to controls, or showing persistent hunching, labored breathing, or other signs of severe distress, were immediately euthanized to minimize pain and suffering.

### Cell culture

To obtain bone marrow cells (BMCs), femurs and tibias were crushed in PBS supplemented with 2% heat-inactivated fetal bovine serum (FBS). After cells were filtered through a 40-μm mesh, red blood cells were hemolyzed with ACK lysing buffer (GIBCO, NY, USA). The cells were cultured in RPMI 1640 medium supplemented with 10% FBS, 100 units/ml penicillin, and 100 μg/ml streptomycin.

Human umbilical vein endothelial cells (HUVECs, product code C-12208) were obtained from PromoCell (Sickingenstr, Heidelberg, Germany). They were supplied from pooled donors. HUVECs were cultivated in Endothelial Cell Growth Medium 2 (PromoCell, Sickingenstr, Heidelberg, Germany). The medium contains 2% FBS. HUVECs were incubated for 4 h in the presence or absence of α-toxin and the candidate compounds. Cell viability was measured using CCK-8. The median effective concentrations ($EC_{50}$) of micafungin and caspofungin for inhibiting α-toxin-mediated cell death were calculated using GraphPad Prism (9.5.1). A LDH

leakage assay was performed using Cytotoxicity LDH Assay Kit-WST according to the manufacturer's protocol.

## Bacterial culture and infection

Bacterial culture and infection were performed as previously described[12]. In brief, *C. perfringens* strain 13 was grown in tryptone, glucose, and yeast extract (TGY) medium under anaerobic conditions at 37 °C. Exponentially growing bacteria were harvested, washed, re-suspended in TGY medium, and injected into the femoral muscle of mice. Immediately after administering bacteria, caspofungin and/or clindamycin diluted in PBS was intraperitoneally administered. To quantify CFUs, residual bacteria were serially diluted, plated on brain heart infusion agar plates, and cultured anaerobically at 37 °C.

In the experiment assessing mouse survival, exponentially growing bacteria ($9 \times 10^8$ CFU) were harvested, washed, re-suspended in TGY medium, and injected into the femoral muscle of mice. Immediately after administering bacteria, 1 mg of caspofungin and/or 2 mg of clindamycin diluted in PBS was intraperitoneally administered, and the survival of mice was monitored. The experimental unit was a cage of animals. The exact number of mice per group and the total number of mice per experiment are provided in the figure and figure legend. No a priori sample size calculation was performed. Sample sizes were chosen based on previous experience with similar experiments and expected variability. No specific exclusion criteria were established, and no data points or animals were excluded from the analysis. The experiment was conducted by investigators who were aware of the group allocations, while outcome assessment was performed by a separate evaluator who was blinded to the group assignments. Humane endpoints were established such that animals exhibiting more than 20% body weight loss compared to controls, or showing persistent hunching, labored breathing, or other signs of severe distress, were immediately euthanized to minimize pain and suffering.

## Flow cytometry analysis

Cultured BMCs were labeled with antibodies diluted 1:20 in PBS containing 2% FBS after blocking Fc receptors with purified rat anti-mouse CD16/CD32 at 1:100. Labeled cells were assessed using Guava easyCyte (Millipore, MA, USA), and data were analyzed by FlowJo software (Tree Star, OR, USA). Cells stained with propidium iodide were excluded from analyses.

## ELISA

HUVECs were cultured in Endothelial Cell Growth Medium 2 in accordance with the manufacturer's protocol. After the treatment of cells with α-toxin, peptidoglycan (PGN), and the test compounds, culture supernatants were harvested and interleukin-6 levels were measured using a human IL-6 Quantikine ELISA kit (R&D Systems, MN, USA).

## Myotube morphology analysis

Exponentially growing bacteria ($1 \times 10^8$ CFU) were harvested, washed, re-suspended in TGY medium, and injected into the femoral muscle of mice. The experimental unit was a cage of animals. Three mice per condition were used across two experiments. *C. perfringens*-infected muscles were isolated 24 h after infection. Isolated tissues were fixed in 4% paraformaldehyde and embedded in paraffin. Paraffin sections were cut from the tissue and stained with hematoxylin and eosin to visualize muscle fibers. Images of muscle fibers were taken using a digital camera, and the diameters of muscle fibers were measured using Image J software. The diameters of 300 muscle fibers were assessed for each condition.

## Docking simulations

PDB structure file of *C. perfringens* α-toxin (PDBID: 1qm6) was downloaded from the RCSB repository and the PDB file was processed with UCFS Chimera to create pdbqt file (https://www.rcsb.org)[20,21]. The 3D data of micafungin and caspofungin were generated by MoleView and corrected, edited, and conformation-searched by Avogadro, and converted to pdbqt files using UCFS Chimera respectively (https://molview.org)[22,23].

Preliminary docking simulations of micafungin, caspofungin and phosphatidylcholine with α-toxin were performed using AutoDock vina (1.1.2) to discover potential binding sites in α-toxin[24,25]. Then, the 23 amino acids (asp58, tyr62, leu64, tyr65, gln66, asp67, trp70, asp71, thr74, phe78, ser89, ile90, pro91, asp92, thr93, gln97, lys100, phe101, tyr261, asp293, pro295, lys 304, thr306) in α-toxin were selected from those that are within 3 angstroms of the ligand molecules and have residues that protrude toward the ligands, and two pdbqt files required to set rigid and flexible residues in α-toxin were created by AutoDockTools respectively. The flexible docking simulations using AutoDock vina were performed under the following conditions: center_x = −20, center_y = 95, center_z = 95, size_x = 40, size_y = 25, size_z = 35, energy_range = 3, exhaustiveness = 10, num_modes = 20. Finally, the experimental results were visualized and analyzed using UCFS Chimera. The resulting binding score for the best pose of micafungin was −9.5 and for caspofungin was −6.7.

## Statistics and reproducibility

Drug screening was conducted in a single experiment; however, all subsequent in vitro assays were performed at least three times, and representative results are shown in the figures. In addition, the in vivo experiments were conducted under the same conditions in two independent rounds, and the data were combined into each single figure.

All statistical analyses were performed with Easy R (Saitama Medical Center, Jichi Medical University)[26]. Differences between two groups were evaluated by the two-tailed Student's *t* test. A one-way analysis of variance (ANOVA) followed by Tukey's test was used to evaluate differences among three or more groups. The Log-rank test followed by a Bonferroni multiple comparison analysis was performed to assess the significance of differences in survival. Differences were considered to be significant at $P < 0.05$.

## Results

### Drug screening identifies micafungin and caspofungin as α-toxin inhibitors

We screened a library of 764 FDA-approved drugs for compounds that inhibited the PLC activity of α-toxin by 50% or more, and identified 40 compounds (Fig. 1A). To confirm reproducibility, we measured the effects of individual compounds on PLC activity. The results obtained confirmed that 21 compounds inhibited PLC activity by more than 50% (Fig. 1A and Supplementary Fig. 1). We previously reported that α-toxin-induced endothelial cell death[16]. Therefore, as a secondary screen, we investigated whether these compounds inhibited the α-toxin-induced cell death of HUVECs. The concentration of the compounds used in the secondary screen was set to 50 μM, which exerted the strongest inhibitory effect on PLC activity among the many compounds identified in the initial screen. While the viability of cells treated with α-toxin alone decreased to approximately 20%, α-toxin-induced decreases in cell viability were no longer observed in cells co-treated with micafungin, an antifungal agent, and no other compounds exerted similar effects (Fig. 1B). Several compounds, such as butoconazole nitrate, daunorubicin, and idarubicin, exhibited cytotoxicity on their own, but did not significantly inhibit PLC activity at a concentration of 5 μM; therefore, they were not examined further (Fig. 1B and Supplementary Fig. 1). Caspofungin, a drug with comparable efficacy, inhibited PLC activity in a manner similar to micafungin, although micafungin exhibited a stronger inhibitory effect (Fig. 1C). However, contrary to expectations, the inhibitory effects of micafungin on SMase activity were negligible, while caspofungin enhanced it, suggesting that micafungin preferentially inhibits PLC activity and caspofungin possesses dual properties, namely, suppressing PLC activity and promoting SMase activity (Fig. 1D). To determine whether micafungin and caspofungin directly bind to α-toxin, titration experiments were conducted using circular dichroism (CD) spectroscopy. The results indicated that micafungin and caspofungin bind to the toxin, as evidenced by changes in CD intensity (Supplementary Fig. 2).

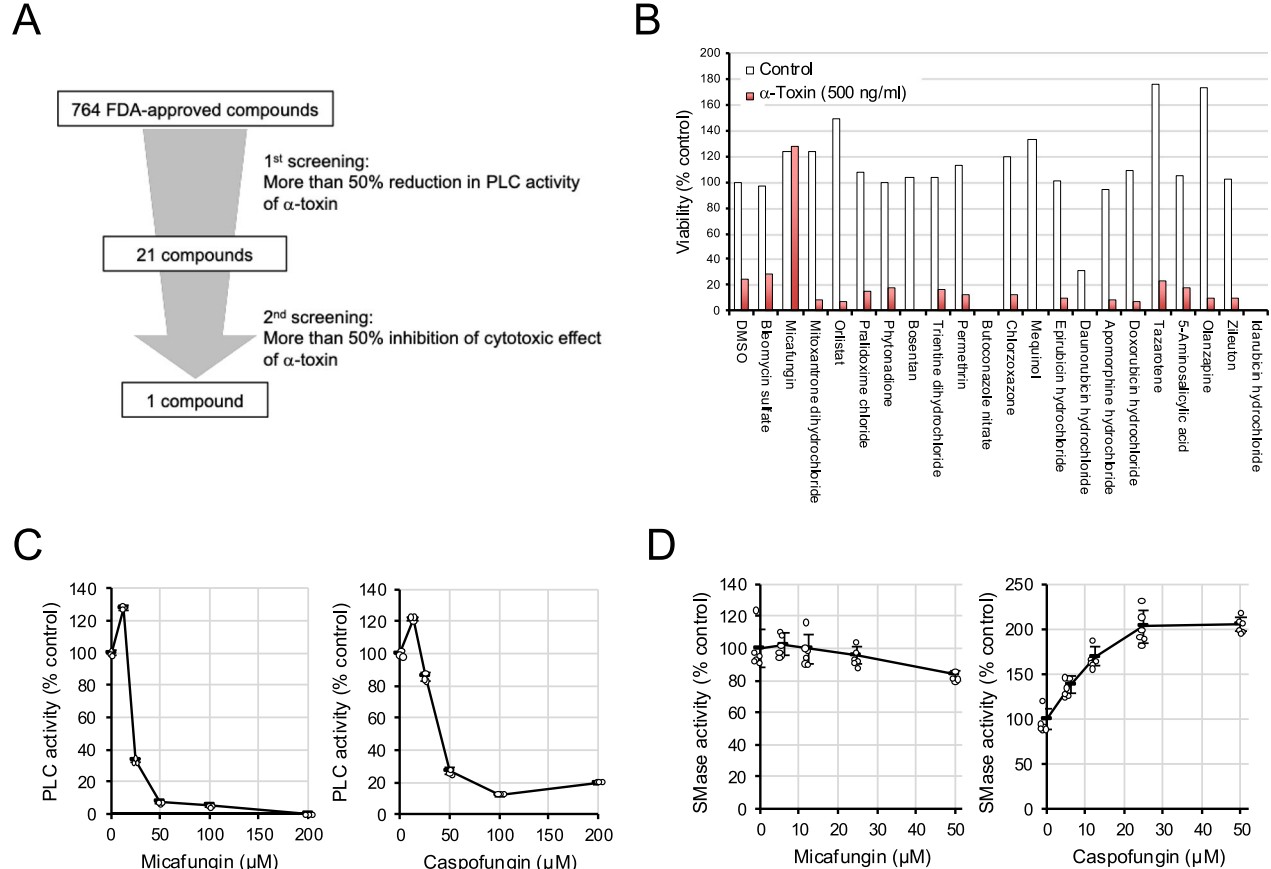

**Fig. 1 | Micafungin and caspofungin inhibit the PLC activity of *C. perfringens* α-toxin. A** Schematic of the screening process. The effects of primarily 50 μM of the test compounds (only one at 10 μM) on the PLC activity of 50 ng/ml α-toxin were measured using the Amplex red phosphatidylcholine-specific phospholipase C assay kit according to the manufacturer's instructions. After one hour of incubation, the measurement was performed by detecting fluorescence intensity with excitation at 550 nm and emission at 580 nm. The screening was performed once, and the reproducibility of the selected candidate compounds was confirmed in the experiment shown in Supplementary Fig. 1. **B** HUVECs were cultured for 4 h in the presence or absence of α-toxin and 50 μM of the candidate compounds. Cell viability was measured using Cell Counting Kit-8 and expressed relative to the control. The screening was performed once, and the reproducibility of the selected candidate compound was confirmed in the experiment shown in Fig. 3A. **C** The effects of 12.5, 25, 50, 100, or 200 μM micafungin or caspofungin on PLC activity were measured

using the Amplex red phosphatidylcholine-specific phospholipase C assay kit according to the manufacturer's instructions. **D** The effects of 6.25, 12.5, 25, or 50 μM micafungin or caspofungin on SMase activity were measured using Amplex red sphingomyelinase assay kit according to the manufacturer's instructions. The measurement was performed by detecting fluorescence intensity with excitation at 550 nm and emission at 580 nm. **C, D** A one-way ANOVA was employed to assess significance. **C** After correction for multiple comparisons, comparisons between the control and each of the five micafungin-treated and five caspofungin-treated data points resulted in adjusted $p$ values of <0.001. **D** Comparisons between the control and each of the four micafungin-treated data points yielded adjusted $p$ values of 0.981, 1.000, 0.897, and 0.038, respectively, whereas all four caspofungin-treated data points showed adjusted $p$ values of <0.001. Values are the mean ± standard deviation for panels (**C**) ($n = 3$) and (**D**) ($n = 4$–6).

## Docking simulations reveal distinct binding modes of micafungin and caspofungin to α-toxin

To clarify the cause of the difference in inhibitory activity between micafungin and caspofungin against α-toxin, docking simulations of these compounds with α-toxin (PDBID: 1qm6) were performed. The side chain of micafungin is stereo chemically rigid and linear than that of caspofungin and is expected to bind to the elongated space of α-toxin in a deeply piercing manner (Fig. 2A, yellow). On the other hand, the side chain of caspofungin is flexible and inductively fits into the shallow part of the receptor space (Fig. 2A, orange). Due to these differences in binding state, the location of the cyclic peptide moiety exposed on α-toxin surface differs between micafungin and caspofungin (Fig. 2B–D). Additionally, the binding scores obtained from these simulation experiments are better for micafungin (−9.5) than for caspofungin (−6.7), which correlates with the inhibitory activities. These differences are presumably responsible for the difference in inhibitory activities for α-toxin between micafungin and caspofungin. Moreover, these compounds bind to sites that partially overlap with the binding sites of phosphatidylcholine, especially the fatty acid moiety that

interacts with α-toxin (Fig. 2A, E). The results suggest that they may inhibit α-toxin by competing with its natural substrate.

Next, interactions between micafungin or caspofungin and α-toxin were analyzed using PLIP (Protein-Ligand Interaction Profiler). Seven hydrophobic interactions, seven hydrogen bonds, one π–π stacking interaction, and one metal coordination were detected between micafungin and α-toxin (Supplementary Fig. 3A). In contrast, one hydrogen bond and ten hydrophobic interactions were identified between caspofungin and α-toxin (Supplementary Fig. 3B). Focusing on the difference in the number of hydrogen bonds, which represent relatively strong intermolecular interactions, micafungin was found to form more hydrogen bonds with α-toxin than caspofungin. This property is thought to be the primary factor contributing to the high affinity of micafungin for α-toxin.

## Caspofungin and micafungin suppress the toxicity of α-toxin

Toxin-induced endothelial cell death was inhibited by micafungin at concentrations of 1.6 μM and higher and by caspofungin at 6.3 μM or higher; therefore, micafungin exerted inhibitory effects on cytotoxicity at lower concentrations (Fig. 3A). The $EC_{50}$

**Fig. 2 | Docking simulation for micafungin and caspofungin with *C. perfringens* α-toxin.** Results of docking simulation for micafungin, caspofungin and phosphatidylcholine with *C. perfringens* α-toxin (PDBID: 1qm6) performed using AutoDock vina are shown. **A** Micafungin (yellow) and caspofungin (orange) bind to the same cavity of α-toxin (light blue), but there is a difference in the location of the respective cyclic peptide parts covering the α-toxin surface. **B** Details of the binding state of the α-toxin and micafungin. The rigid side chain of micafungin penetrates deep into the cavity of α-toxin. **C** Details of the binding state of the α-toxin and caspofungin. The flexible side chain of caspofungin binds inducibly to the shallow part of the cavity in α-toxin. **D** Differences in exposure of micafungin (left, yellow) and caspofungin (right, orange) to α-toxin surfaces. **E** Docking simulation for phosphatidylcholine with α-toxin. Overlay of the top 10 predicted conformers is shown.

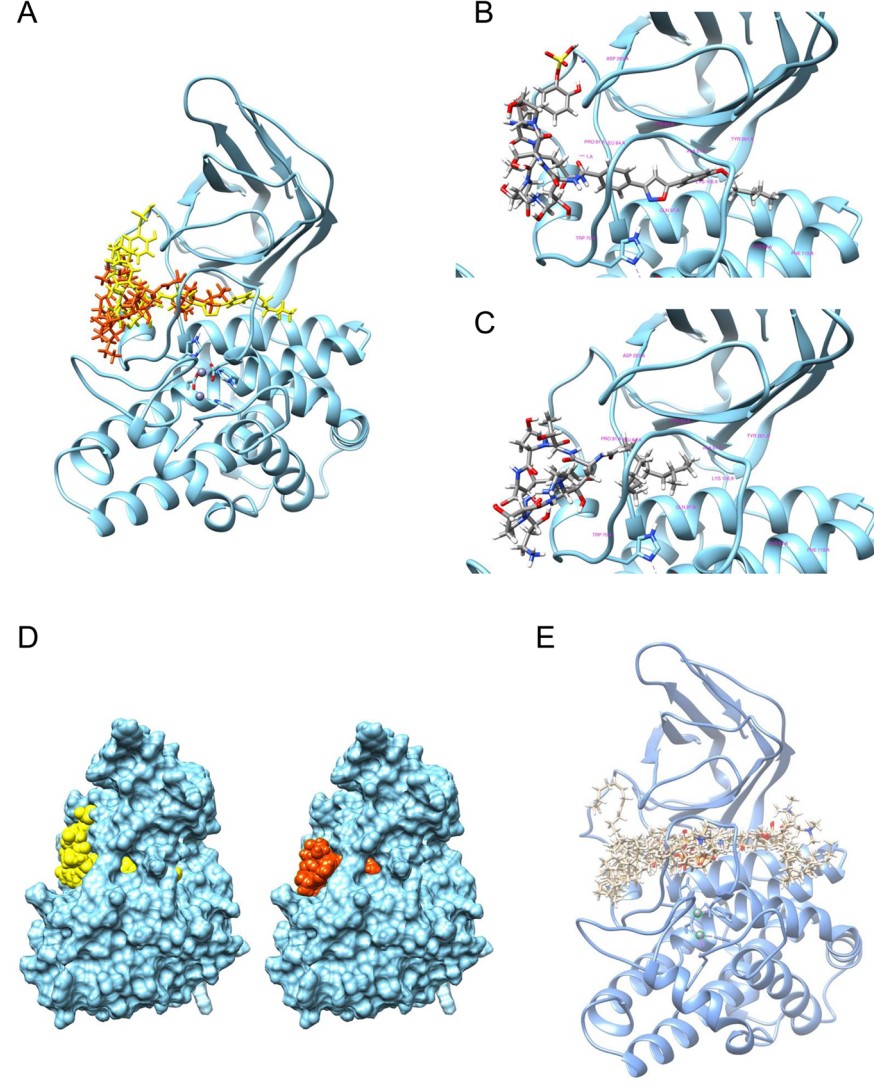

of micafungin and caspofungin were calculated to be 5.6 μM and 22.6 μM, respectively. The inhibition of cytotoxicity by these compounds was also confirmed by a lactate dehydrogenase (LDH) leakage assay (Fig. 3B). These results indicate that α-toxin-induced endothelial cell death is dependent on PLC activity. The inhibitory effects of these compounds were observed at concentrations lower than those that inhibited PLC activity. This may be due to differences in the compositions of solutions in experimental systems and between the cellular substrates and lipids in the PLC assay kit used in this study. α-Toxin exhibits biological activities other than cytotoxicity, such as increasing the expression of CD11b in neutrophils and cytokine production in endothelial cells, which are related to disease progression[11,27]. Micafungin and caspofungin both suppressed interleukin-6 production by HUVECs stimulated with peptidoglycan and α-toxin, demonstrating the involvement of PLC activity in the acceleration of cytokine production by α-toxin (Fig. 3C). On the other hand, micafungin, but not caspofungin, inhibited α-toxin-induced increases in CD11b expression in neutrophils (Fig. 3D). This difference may be due to the different effects of these compounds on SMase activity. In other words, the α-toxin-induced increase in CD11b expression is, at least in part, mediated by SMase activity, which is supported by previous findings showing that *Bacillus cereus* SMase exerted similar effects[11]. We then investigated whether micafungin and caspofungin reduced the lethality of α-toxin in mice. In control mice intraperitoneally injected with α-toxin alone, more than 80% of mice died within 24 h, whereas ~76% of mice simultaneously administered α-toxin and caspofungin survived for more than one week (Fig. 3E). This result indicates the involvement of PLC activity in the lethal activity of α-toxin in mice. Furthermore, contrary to expectations, micafungin did not affect the lethal activity of the toxin. As shown in Supplementary Fig. 4, neither

micafungin nor caspofungin suppressed the lethal activity of α-toxin at the lower dose (100 μg per mouse), in contrast to the higher-dose condition shown in Fig. 3E. Thus, under the conditions used in this study, a dose of 300 μg per mouse is required to suppress the lethal activity.

### Therapeutic potential of caspofungin in a mouse model of *C. perfringens*-induced gas gangrene

To examine the efficacy of caspofungin against gas gangrene, it was intraperitoneally administered to mice infected with *C. perfringens* type A. Both caspofungin and clindamycin, a drug commonly used to treat gas gangrene, significantly improved survival in mice infected with *C. perfringens* (Fig. 4A). The therapeutic effects of the two drugs appear to differ: while caspofungin extended the survival time, it did not improve the overall survival rate, in contrast to clindamycin, which markedly improved the survival rate. This difference in the effects of these compounds may be attributed to their mechanisms of action. Caspofungin only targets α-toxin and, thus, exerts a temporary therapeutic effect, whereas clindamycin, an antibiotic, kills all bacteria, which completely removes infection in this experimental system. Caspofungin also trended to improve survival in clindamycin-treated mice, although not statistically significant (Fig. 4A). In mice infected with *C. perfringens*, the diameter of myotubes in infected muscle decreased and muscle damage was observed. As shown in Fig. 4B, C, caspofungin and clindamycin both inhibited decreases in muscle diameters caused by *C. perfringens* infection, and a synergistic therapeutic effect was observed in mice co-administered these compounds. Collectively, the

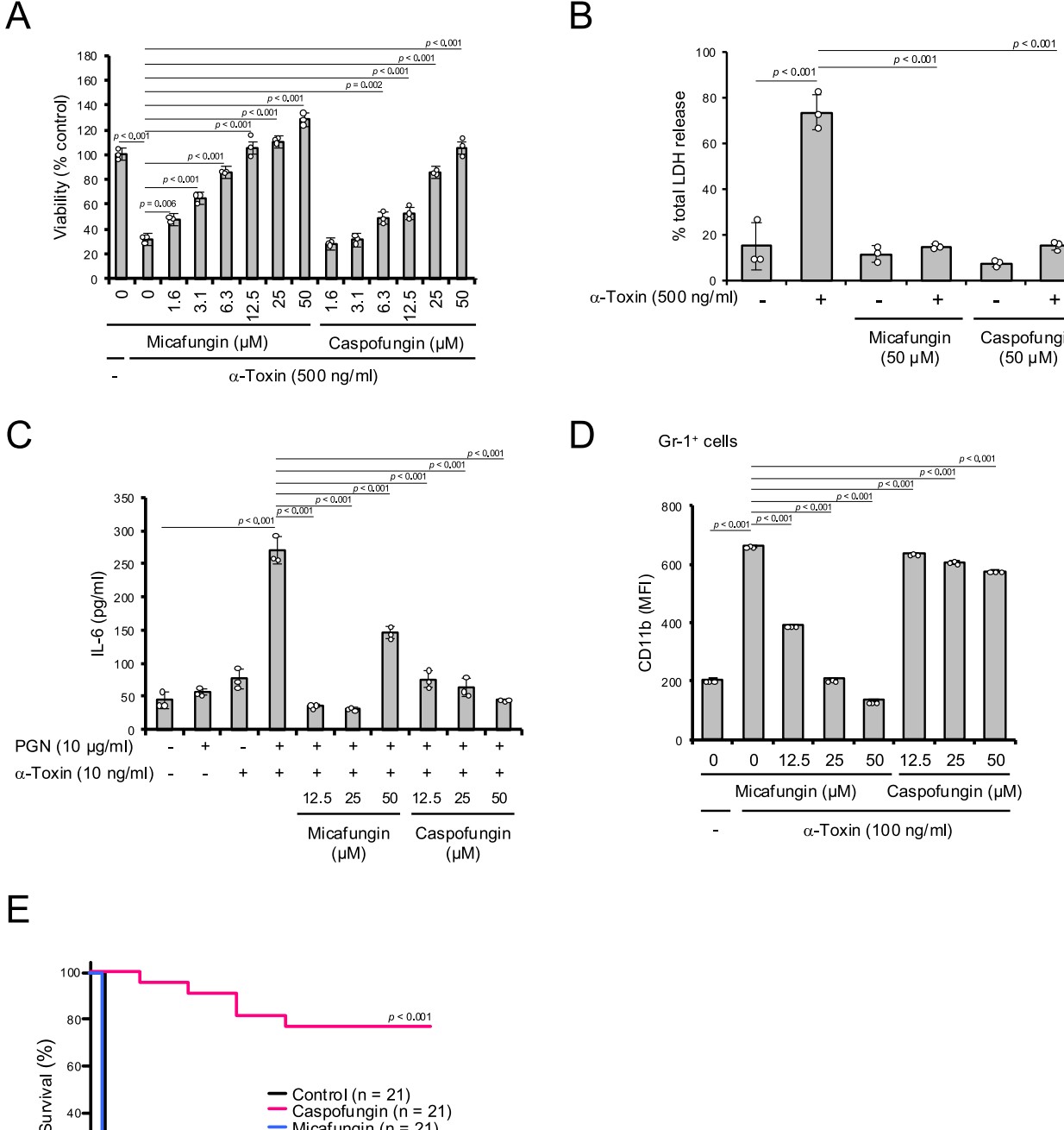

**Fig. 3 | Micafungin and caspofungin inhibited the cytotoxicity of α-toxin.**
**A** HUVECs were cultured for 4 h in the presence or absence of 500 ng/ml α-toxin and the indicated concentrations of micafungin or caspofungin. Cell viability was measured using Cell Counting Kit-8 and expressed relative to the control.
**B** HUVECs were cultured in the presence or absence of 500 ng/ml α-toxin and 50 μM micafungin or caspofungin, and a LDH leakage assay was performed using Cytotoxicity LDH Assay Kit-WST according to the manufacturer's protocol. LDH release was expressed relative to the control. **C** HUVECs were cultured for 24 h in the presence or absence of 10 ng/ml α-toxin, 10 μg/ml peptidoglycan (PGN), and the indicated concentrations of micafungin or caspofungin. Interleukin-6 (IL-6) levels in the culture medium were measured using a human IL-6 Quantikine ELISA kit according to the manufacturer's protocol. **D** Bone marrow cells were cultured for 3 h

in the presence or absence of 100 ng/ml α-toxin and the indicated concentrations of micafungin or caspofungin, and the cells were labeled with antibodies diluted in PBS containing 2% FBS after blocking Fc-receptors with purified rat anti-mouse CD16/CD32. To quantify the amount of CD11b, a flow cytometry analysis was performed using Guava easyCyte. The mean fluorescence intensities of CD11b in Gr-1[+] cells are shown. **E** C57BL/6J mice were injected intraperitoneally with 600 ng of α-toxin and 300 μg of micafungin or caspofungin, which had been pre-mixed and diluted in PBS. The survival of mice was monitored, and Kaplan–Meier survival curves are shown. The total number of mice used in the experiments was 63. Median survival (hours): Control, 6; caspofungin, not reached; micafungin, 6. A one-way ANOVA (**A–D**) or the Log-rank test (**E**) was employed to assess significance. Values are the mean ± standard deviation for panels (**A–D**) ($n = 3$).

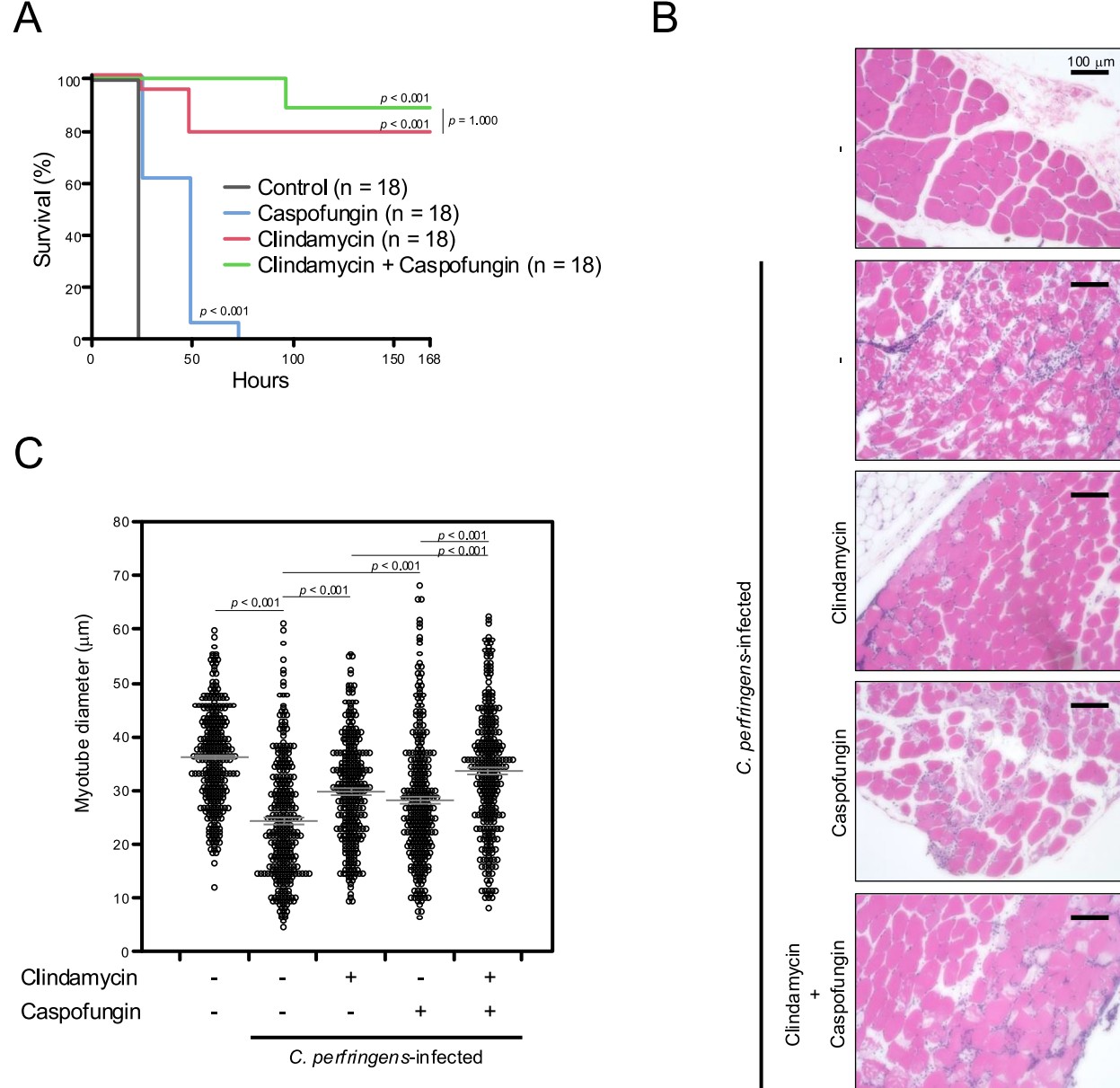

**Fig. 4 | Caspofungin improved survival and muscle injury in mice after *C. perfringens* infection.** *C. perfringens* strain 13 was grown in tryptone, glucose, and yeast extract (TGY) medium under anaerobic conditions at 37 °C. Exponentially growing bacteria ($9 \times 10^8$ CFU for panel (**A**) or $1 \times 10^8$ CFU for panels (**B**, **C**) were harvested, washed, re-suspended in TGY medium, and injected into the femoral muscle of mice. Immediately after administering bacteria, 1 mg of caspofungin and/or 2 mg of clindamycin diluted in PBS was intraperitoneally administered. **A** The survival of mice was monitored, and Kaplan–Meier survival curves are shown. The results of two independent experiments were compiled to generate the figure. The total number of mice used in the experiments was 72. Median survival (hours): control, 24; caspofungin, 48; clindamycin, not reached; clindamycin plus caspofungin, not reached. **B**, **C** *C. perfringens*-infected muscles were isolated 24 h after infection. Isolated tissues were fixed in 4% paraformaldehyde and embedded in paraffin. Paraffin sections were cut from the tissue and stained with hematoxylin and eosin (H&E) to visualize muscle fibers. Images of muscle fibers were taken using a digital camera. Representative H&E-stained sections are shown (**B**), and 300 muscle fibers were measured using Image J software (**C**). The Log-rank test (**A**) or a one-way ANOVA (**C**) was employed to assess significance. Values are the mean ± standard error.

present results suggest caspofungin as a treatment for *C. perfringens*-induced gas gangrene, providing a potential therapeutic benefit when added to current treatments.

## Discussion

In the present study, we screened a library of 764 FDA-approved drugs for compounds that inhibited PLC activity of α-toxin and identified micafungin. Caspofungin, a drug with comparable efficacy, inhibited PLC activity in a manner similar to micafungin. Contrary to expectations, the inhibitory effects of micafungin on SMase activity were negligible, while caspofungin enhanced it, suggesting that micafungin preferentially inhibits PLC activity and caspofungin possesses dual properties, namely, suppressing PLC activity and promoting SMase activity. These different effects on enzymatic activities may be attributed to the compound binding to sites involved in the lipid recognition of the toxin. Moreover, since these compounds exerted different effects on PLC and SMase activities, their use will provide more detailed insights into the molecular mechanisms by which the biochemical activity of α-toxin contributes to the development of gas gangrene.

Micafungin and caspofungin inhibit fungal cell wall synthesis. Specifically, these agents inhibit FKS, a β-1,3-glucan synthase[28]. It might be an

intriguing question whether the structural features of the binding site for micafungin and caspofungin on α-toxin, which we identified in this study, share similarity with those of FKS. Such structural similarity-if present-would increase the plausibility that these compounds were selected in our screening. FKS is an approximately 200-kDa membrane-spanning protein with low sequence similarity to other known glycosyltransferases (GTs). Its crystallographic analysis has been challenging, and although FKS belongs to the GT48 family, no structures were available for any of its members. Recently, however, the structure of *Saccharomyces cerevisiae* FKS1 was determined by cryo-electron microscopy[29]. Nevertheless, the binding sites of micafungin and caspofungin on FKS1 have not yet been identified. For this reason, we were unable to compare the structural similarity of compound-binding sites between α-toxin and FKS1. We believe that the binding-mode information of α-toxin and the compounds elucidated in this study may provide valuable clues toward identifying the binding sites of the compounds on FKS1 and potentially contribute to further advances in FKS1 research.

In this study, we demonstrated that caspofungin decreases the lethality of α-toxin in mice; however, micafungin does not. The pharmacokinetics of these compounds in humans, such as their blood half-lives and maximum blood concentrations, are similar[30,31], but the plasma protein binding rate of micafungin is slightly higher than that of caspofungin (97% for caspofungin and 99% for micafungin)[32]. Although the absolute difference between these values is small, it could still lead to a significant difference in efficacy if the compounds lose their inhibitory activity against α-toxin when bound to plasma proteins. In other words, if only the unbound fraction is active, plasma protein binding may critically influence their effectiveness. In addition, unknown factors may contribute to the differences in efficacy between micafungin and caspofungin, and further investigation is needed to clarify these mechanisms.

A previous study reported that the maximum blood concentration of micafungin in humans was ~6 μM[31]. The concentration of the compound that suppressed PLC activity in our experimental system was 25 μM, which is several-fold higher than the maximum blood concentration in humans. On the other hand, caspofungin inhibited toxin-induced endothelial cell death at concentrations of 6.3 μM or higher, which is equivalent to the maximum blood concentration in humans. Therefore, the concentrations of caspofungin set in the present study were close to clinical values, indicating its potential in the treatment of gas gangrene with improved administration schedules, such as local and single high-dose administration protocols.

To date, in cases of gas gangrene caused by *C. perfringens* type A, early surgical debridement and administration of antimicrobial agents have been considered the most critical factors for improving survival, and these approaches constitute the standard treatment[3]. In Japan, an equine-derived antitoxin preparation produced using an α-toxin toxoid has been approved; thus, an approved therapeutic agent targeting this toxin already exists[33]. However, because this antitoxin is managed as a national stockpile, its rapid use is limited, making it difficult to respond to the rapid progression of gas gangrene. In addition, equine-derived antitoxins carry a risk of undesirable immune reactions, which further restricts their clinical use. Our findings may therefore highlight the importance of complementing existing α-toxin-targeted therapy. Specifically, caspofungin is a widely available pharmaceutical agent and could be rapidly administered to patients with gas gangrene without concerns regarding anaphylaxis to heterologous components, thereby potentially advancing therapeutic strategies targeting this toxin.

The market size for many infectious diseases caused primarily by bacterial toxins is not large and, thus, it is often difficult from a market economic perspective to develop medicines targeting these toxins. Among the pharmaceuticals in clinical use, very few drugs target bacterial toxins other than toxoids and antitoxins. In the present study, we screened a library of FDA-approved drugs and found that caspofungin may be beneficial for the treatment of *C. perfringens*-induced gas gangrene by inhibiting α-toxin. Screening using an approved drug library may be useful for reducing development costs and is considered suitable for developing therapeutic drugs targeting other bacterial toxins. The present study provides conceptual advances in the treatment of bacterial infections, and will contribute to the discovery of newly classified therapeutic drugs targeting bacterial toxins.

## Data availability
The source data underlying Figs. 1B–D, 3A–E, 4A, C, Supplementary Figs. 1, 2, and 4 are provided in Supplementary Data 1. The file includes all numerical values used to generate the graphs and charts presented in the main figures. *C. perfringens* strain 13 is available from the corresponding author upon reasonable request. No other unique materials were generated by studies described in manuscript.

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

## Acknowledgements

The authors thank Takahiro Takabatake for his technical assistance. This study was supported by the Japan Agency for Medical Research and Development (AMED) under grant numbers JP22fk0108635, JP23fk0108635, JP24fk0108635 and JP25fk0108728 to M.T., and a grant-in-aid of the Fugaku Trust for Medical Research to T.Y.

## Author contributions

M.T. conceptualized and designed the study, performed experiments, and wrote the manuscript; Y.H. contributed to murine infection studies. T.I., Y.S., and Y.K. performed the data analysis. Y.K. and H.I. made contributions to the docking simulation studies. K.M., K.N., and Y.T. conducted titration experiments monitored by CD spectroscopy. T.Y., H.I., and M.N. supervised experiments.

## Competing interests

The authors declare no competing interests.

## Additional information

[1]Department of Microbiology and Immunology, Faculty of Pharmacy, Juntendo University, Chiba, Japan. [2]Department of Microbiology, Faculty of Pharmaceutical Sciences, Tokushima Bunri University, Tokushima, Japan. [3]Department of Pharmacy, Faculty of Pharmaceutical Sciences, Nagasaki International University, Nagasaki, Japan. [4]Chemistry of Functional Molecule, Faculty of Pharmaceutical Sciences, Tokushima Bunri University, Tokushima, Japan. [5]Laboratory of Analytical Chemistry, Faculty of Pharmaceutical Sciences, Tokushima Bunri University, Tokushima, Japan. ✉e-mail: m.takehara.du@juntendo.ac.jp

