## [Transparent Peer Review file · Communications Medicine]

Repurposing caspofungin as a small-molecule inhibitor of *Clostridium perfringens* α -Toxin for treatment of gas gangrene

Corresponding Author: Dr Masaya Takehara

Version 0:

Reviewer comments:

Reviewer #1

(Remarks to the Author)

In their manuscript entitled "Repurposing caspofungin as a small-molecule inhibitor of *Clostridium perfringens* α -Toxin for treatment of gas gangrene", the authors screened an FDA-approved drug library that contained 764 compounds to identify pharmacological inhibitors of the *Clostridium perfringens* α -Toxin, which is a medically relevant bacterial protein toxin that harbours a phospholipase (PLC) activity, and the causative agent for the severe post-traumatic disease gas gangrene. By screening the compounds in vitro for inhibition of the PLC activity, they found 21 inhibitors. Next, they tested these compounds in a cell-based approach whether they can protect human HUVEC cells from the cytotoxic action of α -Toxin. Here, only micafungin, an anti-fungal drug showed toxin-neutralizing activity. Therefore, the authors included the comparable drug caspofungin into the next steps of investigation. Both anti-fungal drugs prevented the toxin-induced activation of human neutrophils and the production of cytokines from HUVEC cells in vitro. In the last step, both compounds were tested for toxin-neutralizing activities in vivo in a mouse model where mice were either challenged with the α -toxin, or with *C. perfringens* type A, which produces that toxin. In both approaches, caspofungin but not micafungin showed protective effects, i.e. it significantly reduced the lethality caused by α -toxin, and had positive effects on the survival and mitigated muscle damage of infected mice. In addition, the authors provide a model by docking simulations, showing binding of caspofungin and micafungin to the toxin.

Overall, this is a medically and scientifically relevant topic, the methodology is sound and the manuscript well written. The steps of investigation from in vitro via cell-based assays to in vivo models is plausible and the results are overall convincing. However, there are some points that should be addressed:

1. Based on the docking model, the binding of caspofungin and/or micacystin to α -toxin should be confirmed in vitro by ITC, SPR or other appropriate methods. This would also provide quantitative data about the affinity of that interaction.
2. Both anti-fungal drugs inhibit an enzyme that is crucial for cell wall synthesis (glucan synthesis) in the fungi. This should be at least shortly discussed. Is there any structural relation between these fungal enzymes and PLC?
3. Regarding the in vivo approach, it might be beneficial to apply the caspofungin not only at the beginning of the observation course, but frequently, to achieve a more robust or longer lasting toxin inhibition and thereby an even better outcome of the intoxicated or infected mice. Has this been considered?

Reviewer #2

(Remarks to the Author)

This study identified potential drugs that can inhibit the toxicity of *Clostridium perfringens* alpha-toxin, which is significant. However, the authors have conducted insufficient systematic research and in-depth exploration on this finding, with somewhat inadequate evidence, which is concerning.

1. The methodological description of the mouse challenge experiment is missing.
2. It is suggested that the Results and Discussion sections be divided into several hierarchical parts with headings added to enhance readability.
3. The authors appear to have conducted insufficient in-depth analysis of the docking results, failing to clarify the specific differences in the binding of the two drugs—for example, which amino acids and chemical bonds play key roles.

4. There is a lack of data on the protective effects of drugs at different concentrations in mice, and supplementation is recommended.

5. The strategy of "screening using an approved drug library to reduce development costs" is commendable, as it holds great significance for the development of therapeutic drugs targeting toxins with small market sizes. The authors should conduct in-depth discussions on whether the minor effects of inhibitors targeting a single toxin are practically meaningful, given that antibiotics have good therapeutic effects and *Clostridium perfringens* can produce multiple toxins.

Version 1:

Reviewer comments:

Reviewer #1

(Remarks to the Author)

My concerns were adequately addressed by the authors in their revised manuscript. I do not have further points.

Reviewer #2

(Remarks to the Author)

The authors have addressed and revised the issues raised in the first review, and no further questions remain.

Reviewers' comments:

Reviewer #1 (Remarks to the Author):

In their manuscript entitled "Repurposing caspofungin as a small-molecule inhibitor of Clostridium perfringens α -Toxin for treatment of gas gangrene", the authors screened an FDA-approved drug library that contained 764 compounds to identify pharmacological inhibitors of the Clostridium perfringens α -Toxin, which is a medical relevant bacterial protein toxin that harbors a phospholipase (PLC) activity, and the causative agent for the severe post-traumatic disease gas gangrene. By screening the compounds in vitro for inhibition of the PLC activity, they found 21 inhibitors. Next, they tested these compounds in a cell-based approach whether they can protect human HUVEC cells from the cytotoxic action of α -Toxin. Here, only micafungin, an anti-fungal drug showed toxin-neutralizing activity. Therefore, the authors included the comparable drug caspofungin into the next steps of investigation. Both anti-fungal drugs prevented the toxin-induced activation of human neutrophils and the production of cytokines from HUVEC cells in vitro. In the last step, both compounds were tested for toxin-neutralizing activities in vivo in a mouse model where mice were either challenged with the α -toxin, or with C. perfringens type A, which produces that toxin. In both approaches, caspofungin but not micafungin showed protective effects, i.e. it significantly reduced the lethality caused by α -toxin, and had positive effects on the survival and mitigated muscle damage of infected mice. In addition, the authors provide a model by docking simulations, showing binding of caspofungin and micafungin to the toxin.

Overall, this is a medically and scientifically relevant topic, the methodology is sound and the manuscript well written. The steps of investigation from in vitro via cell-based assays to in vivo models is plausible and the results are overall convincing. However, there are some points that should be addressed:

Response: Thank you very much for the positive evaluation of our submitted manuscript. We have revised the paper in response to the reviewer's comments, and we would be grateful if you could kindly proceed with the review.

1. Based on the docking model, the binding of caspofungin and/or micacystin to α -toxin should be confirmed in vitro by ITC, SPR or other appropriate methods. This would also provide quantitative data about the affinity of that interaction.

Response: Thank you very much for your valuable comments. We also consider the reviewer's suggestion to confirm the binding of the compound to the toxin *in vitro* to be a very important issue. Unfortunately, we do not have access to an experimental setup for ITC or SPR; therefore, as an alternative approach, we

performed titration experiments using circular dichroism (CD) spectroscopy as shown in Supplementary Fig. 2. In the experiments, micafungin was successfully analyzed by CD under the conditions of 2.2 μM α -toxin at 37 °C. In contrast, caspofungin could not be analyzed under the same conditions, as the CD signal became saturated at substoichiometric drug concentrations for reasons that remain unclear, yielding an uninterpretable titration curve. Therefore, caspofungin was measured under alternative conditions (1.1 μM α -toxin at 15 °C), which produced a reliable titration curve, and these data were used for subsequent analysis. Although direct comparison of the K_d values was not possible because of the difference in experimental temperatures, these results demonstrate that micafungin and caspofungin bind to the toxin, as evidenced by changes in CD intensity. The results have been described in lines 290-294.

2. Both anti-fungal drugs inhibit an enzyme that is crucial for cell wall synthesis (glucan synthesis) in the fungi. This should be at least shortly discussed. Is there any structural relation between these fungal enzymes and PLC?

Response: Thank you very much for your insightful comments. As the reviewer pointed out, micafungin and caspofungin inhibit fungal cell wall synthesis. Specifically, these agents inhibit FKS, a β -1,3-glucan synthase. As suggested by the reviewer, it is indeed an intriguing question whether the structural features of the binding site for micafungin and caspofungin on α -toxin, which we identified in this study, share similarity with those of FKS. Such structural similarity-if present-would increase the plausibility that these compounds were selected in our screening. FKS is an approximately 200-kDa membrane-spanning protein with low sequence similarity to other known glycosyltransferases (GTs). Its crystallographic analysis has been challenging, and although FKS belongs to the GT48 family, no structures were available for any of its members. Recently, however, the structure of *Saccharomyces cerevisiae* FKS1 was determined by cryo-electron microscopy. Nevertheless, the binding sites of micafungin and caspofungin on FKS1 have not yet been identified. For this reason, we were unable to compare the structural similarity of compound-binding sites between α -toxin and FKS1. We believe that the binding-mode information of α -toxin and the compounds elucidated in this study may provide valuable clues toward identifying the binding sites of the compounds on FKS1 and potentially contribute to further advances in FKS1 research. We have added this discussion to the revised manuscript in lines 391-406.

3. Regarding the in vivo approach, it might be beneficial to apply the caspofungin not only at the beginning of the observation course, but frequently, to achieve a more robust or longer lasting toxin inhibition and thereby an even better outcome of the intoxicated or infected mice. Has this been considered?

Thank you for your comments. We also recognize that the reviewer's point is highly essential and important. In the initial version of the manuscript, we presented results obtained by administering a single 1-mg dose of caspofungin immediately after infection with *C. perfringens*. Although this administration resulted in a statistically significant prolongation of survival, the difference from the untreated control group was limited. In our preliminary investigations, we tested a protocol in which 1 mg of caspofungin was administered every 12 hours starting immediately after infection with *C. perfringens*. Because caspofungin is expensive, the number of mice we were able to test was limited. Nevertheless, multiple dosing did not produce a dramatic improvement in survival or survival time; instead, we observed phenomena suggestive of drug toxicity (Response Figure 1A). In non-infected mice, frequent administration of 1 mg of caspofungin for five days caused no apparent health problems, and no severe toxicity was detected. However, under lethal *C. perfringens* infection, we cannot rule out the possibility that the infection altered the drug's pharmacokinetics and led to the emergence of caspofungin toxicity. Moreover, although we have not examined this in mice, the half-life of caspofungin in humans is approximately 12 hours, which may lead to excessively high plasma levels when administered frequently. Therefore, we reduced the dose to 300 μ g and changed the protocol to daily administration every 24 hours. Using this protocol, we performed the same experiments; however, no statistically significant differences were observed between the control group and the caspofungin-treated group (Response Figure 1B). A previous report by another group demonstrated that, in mice administered α -toxin, neutralizing nanobody against the toxin must be administered promptly after toxin exposure to achieve therapeutic efficacy (R1). When considered together with our results, these findings suggest that inhibitors of α -toxin may need to be administered as quickly as possible and at sufficiently high doses to be effective. In any case, there remains considerable room for optimizing the caspofungin administration protocol. We believe that further examination—including formulation and routes of administration—will be important, and we plan to focus future studies on these aspects.

Response Figure 1. Study on caspofungin dosing protocol. *C. perfringens* strain 13 was grown in tryptone, glucose, and yeast extract (TGY) medium under anaerobic conditions at 37°C. Exponentially growing bacteria (9×10^8 CFU) were harvested, washed, re-suspended in TGY medium, and injected into the femoral muscle of mice. Immediately after administering bacteria, 1 mg (A) or 300 μ g (B) of caspofungin diluted in phosphate-buffered saline was intraperitoneally administered. Thereafter, the same dose of caspofungin

was repeatedly administered every 12 hours. The survival of mice was monitored, and Kaplan-Meier survival curves are shown. Log-rank tests were employed to assess significance.

Reference for reviewer comments

R1 Jia, Q. *et al.* Preparation and Application of Clostridium perfringens Alpha Toxin Nanobodies. *Vet. Sci.* **11** (2024). <https://doi.org/10.3390/vetsci11080381>

Reviewer #2 (Remarks to the Author):

This study identified potential drugs that can inhibit the toxicity of Clostridium perfringens alpha-toxin, which is significant. However, the authors have conducted insufficient systematic research and in-depth exploration on this finding, with somewhat inadequate evidence, which is concerning.

Response: We sincerely thank the reviewers for their valuable and detailed comments. Their suggestions have been extremely helpful in improving the quality of our manuscript. Please see below for our detailed responses.

1. The methodological description of the mouse challenge experiment is missing.

Response: We appreciate your valuable comment. Additional details regarding the mouse challenge experiments have been incorporated in lines 107-110, 117-121, 156-169, 198-211 and 228-229.

2. It is suggested that the Results and Discussion sections be divided into several hierarchical parts with headings added to enhance readability. [EDITORS NOTE: the discussion section should be separate from the Results section. The discussion section cannot have subheadings, but we encourage subheadings in the Results section]

Response: Thank you very much for your comments. The discussion section has been separated from the Results section, and subheadings have been added to the Results section as shown in lines 269, 296-297, 324 and 356-357.

3. The authors appear to have conducted insufficient in-depth analysis of the docking results, failing to clarify the specific differences in the binding of the two drugs—for example, which amino acids and chemical bonds play key roles.

Response: We greatly appreciate this important comment. The point is a key issue in this manuscript. As shown in Supplementary Fig.3, interactions between micafungin or caspofungin and α -toxin were analyzed using PLIP (Protein-Ligand Interaction Profiler). Seven hydrophobic interactions, seven hydrogen bonds, one π - π stacking interaction, and one metal coordination were detected between micafungin and α -toxin (Supplementary Fig. 3A). In contrast, one hydrogen bond and ten hydrophobic interactions were identified between caspofungin and α -toxin (Supplementary Fig. 3B). Focusing on the difference in the number of hydrogen bonds, which represent relatively strong intermolecular interactions, micafungin was found to form more hydrogen bonds with α -toxin than caspofungin. This property is thought to be the primary factor contributing to the high affinity of micafungin for α -toxin. This result has now been described in lines 314-322.

4. There is a lack of data on the protective effects of drugs at different concentrations in mice, and supplementation is recommended.

Response: Thank you very much for your comment. As shown in Supplementally Fig. 4, neither micafungin nor caspofungin suppressed the lethal activity of α -toxin at the lower dose (100 μ g per mouse) compared to the condition in Fig. 3E. (100 μ g per mouse). Thus, under the conditions used in this study, a dose of 300 μ g per mouse is required to suppress the lethal activity. This result has now been described in lines 351-354. In response to another reviewer's comment, we also examined the dose of caspofungin in a mouse model of *C. perfringens* infection. As shown in Fig. 4A, a single administration of caspofungin at 1 mg per mouse resulted in a significant prolongation of survival. In contrast, when caspofungin was administered at a dose of 300 μ g per mouse every 12 hours, no prolongation of survival was observed (Response Figure 1B). These results indicate that, in *C. perfringens*-infected mice, a dose of 1 mg per mouse is required to suppress lethal activity. Although the *in vivo* kinetics of α -toxin following a single administration differ substantially from those in *C. perfringens*-infected mice, these findings suggest that, *in vivo* in mice, caspofungin is generally required at a dose of approximately 300-1000 μ g per mouse to suppress the lethal activity of α -toxin.

Response Figure 1. Study on caspofungin dosing protocol. *C. perfringens* strain 13 was grown in tryptone, glucose, and yeast extract (TGY) medium under anaerobic conditions at 37°C. Exponentially growing bacteria (9×10^8 CFU) were harvested, washed, re-suspended in TGY medium, and injected into the femoral muscle of mice. Immediately after administering bacteria, 1 mg (A) or 300 µg (B) of caspofungin diluted in phosphate-buffered saline was intraperitoneally administered. Thereafter, the same dose of caspofungin was repeatedly administered every 12 hours. The survival of mice was monitored, and Kaplan-Meier survival curves are shown. Log-rank tests were employed to assess significance.

5. The strategy of "screening using an approved drug library to reduce development costs" is commendable, as it holds great significance for the development of therapeutic drugs targeting toxins with small market sizes. The authors should conduct in-depth discussions on whether the minor effects of inhibitors targeting a single toxin are practically meaningful, given that antibiotics have good therapeutic effects and *Clostridium perfringens* can produce multiple toxins.

Response: Thank you very much for this comment. The point raised by the reviewer is extremely important, as it is directly related to the significance of our findings. To date, in cases of gas gangrene caused by *C. perfringens* type A, early surgical debridement and administration of antimicrobial agents have been considered the most critical factors for improving survival, and these approaches constitute the standard treatment. In Japan, an equine-derived antitoxin preparation produced using an α -toxin toxoid has been approved; thus, an approved therapeutic agent targeting this toxin already exists. However, because this antitoxin is managed as a national stockpile, its rapid use is limited, making it difficult to respond to the rapid progression of gas gangrene. In addition, equine-derived antitoxins carry a risk of undesirable immune reactions, which further restricts their clinical use. Our findings may therefore highlight the importance of complementing existing α -toxin-targeted therapies. Specifically, caspofungin is a widely available pharmaceutical agent and could be rapidly administered to patients with gas gangrene without concerns regarding anaphylaxis to heterologous components, thereby potentially advancing therapeutic strategies targeting this toxin. These considerations have been added in line 427-439.

---

## [Transparent Peer Review file · Communications Medicine]

Repurposing caspofungin as a small-molecule inhibitor of *Clostridium perfringens* α -Toxin for treatment of gas gangrene

Corresponding Author: Dr Masaya Takehara

Version 0:

Reviewer comments:

Reviewer #1

(Remarks to the Author)

In their manuscript entitled "Repurposing caspofungin as a small-molecule inhibitor of *Clostridium perfringens* α -Toxin for treatment of gas gangrene", the authors screened an FDA-approved drug library that contained 764 compounds to identify pharmacological inhibitors of the *Clostridium perfringens* α -Toxin, which is a medically relevant bacterial protein toxin that harbours a phospholipase (PLC) activity, and the causative agent for the severe post-traumatic disease gas gangrene. By screening the compounds *in vitro* for inhibition of the PLC activity, they found 21 inhibitors. Next, they tested these compounds in a cell-based approach whether they can protect human HUVEC cells from the cytotoxic action of α -Toxin. Here, only micafungin, an anti-fungal drug showed toxin-neutralizing activity. Therefore, the authors included the comparable drug caspofungin into the next steps of investigation. Both anti-fungal drugs prevented the toxin-induced activation of human neutrophils and the production of cytokines from HUVEC cells *in vitro*. In the last step, both compounds were tested for toxin-neutralizing activities *in vivo* in a mouse model where mice were either challenged with the α -toxin, or with *C. perfringens* type A, which produces that toxin. In both approaches, caspofungin but not micafungin showed protective effects, i.e. it significantly reduced the lethality caused by α -toxin, and had positive effects on the survival and mitigated muscle damage of infected mice. In addition, the authors provide a model by docking simulations, showing binding of caspofungin and micafungin to the toxin.

Overall, this is a medically and scientifically relevant topic, the methodology is sound and the manuscript well written. The steps of investigation from *in vitro* via cell-based assays to *in vivo* models is plausible and the results are overall convincing. However, there are some points that should be addressed:

1. Based on the docking model, the binding of caspofungin and/or micacystin to α -toxin should be confirmed *in vitro* by ITC, SPR or other appropriate methods. This would also provide quantitative data about the affinity of that interaction.
2. Both anti-fungal drugs inhibit an enzyme that is crucial for cell wall synthesis (glucan synthesis) in the fungi. This should be at least shortly discussed. Is there any structural relation between these fungal enzymes and PLC?
3. Regarding the *in vivo* approach, it might be beneficial to apply the caspofungin not only at the beginning of the observation course, but frequently, to achieve a more robust or longer lasting toxin inhibition and thereby an even better outcome of the intoxicated or infected mice. Has this been considered?

Reviewer #2

(Remarks to the Author)

This study identified potential drugs that can inhibit the toxicity of *Clostridium perfringens* alpha-toxin, which is significant. However, the authors have conducted insufficient systematic research and in-depth exploration on this finding, with somewhat inadequate evidence, which is concerning.

1. The methodological description of the mouse challenge experiment is missing.
2. It is suggested that the Results and Discussion sections be divided into several hierarchical parts with headings added to enhance readability.
3. The authors appear to have conducted insufficient in-depth analysis of the docking results, failing to clarify the specific differences in the binding of the two drugs—for example, which amino acids and chemical bonds play key roles.

4. There is a lack of data on the protective effects of drugs at different concentrations in mice, and supplementation is recommended.

5. The strategy of "screening using an approved drug library to reduce development costs" is commendable, as it holds great significance for the development of therapeutic drugs targeting toxins with small market sizes. The authors should conduct in-depth discussions on whether the minor effects of inhibitors targeting a single toxin are practically meaningful, given that antibiotics have good therapeutic effects and *Clostridium perfringens* can produce multiple toxins.

Version 1:

Reviewer comments:

Reviewer #1

(Remarks to the Author)

My concerns were adequately addressed by the authors in their revised manuscript. I do not have further points.

Reviewer #2

(Remarks to the Author)

The authors have addressed and revised the issues raised in the first review, and no further questions remain.
